# Effects of Heat Treatment and Erosion Particle Size on Erosion Resistance of a Hypereutectic High-Chromium Cast Iron

Alessio Suman and Annalisa Fortini *

Department of Engineering, University of Ferrara, 44122 Ferrara, Italy; alessio.suman@unife.it
* Correspondence: annalisa.fortini@unife.it

**Abstract:** This research addresses the erosive resistance of a hypereutectic high-chromium cast iron subjected to solid particle erosion. The study stems from a specific application of high-chromium cast iron, i.e., the critical surfaces of large industrial fans operating in a cement clinker grinding plant where such damage is a limiting factor for the components' lifespan. A dedicated experimental investigation on the impact of substrate microstructure and erodent particle size on erosion resistance was set. The experimental campaign, conducted on a dedicated test bench per the ASTM G76 standard, comprised the analysis of the as-received, tempered, and destabilized conditions for the cast iron. From a preliminary image analysis of the microstructural features, two diameters of the erodent powder for the erosion tests were defined. The observed erosion rate decreased with the increase in the mean particle diameter of the erodent, indicating more severe erosive conditions for smaller particles. From the analysis of the worn surfaces, it was possible to highlight the involved mechanisms concerning the considered test combinations. For the as-received condition, the erosion rate with the larger mean particle diameter of the erodent decreased three times compared to the smaller one. For the heat-treated conditions, the erosion rate was halved with the larger mean particle diameter of the erodent. The proposed analysis, intended to acquire more insight into the limiting factor for the components' lifespan for erosive wear damage, proved that erosion resistance is not dependent on the material's hardness. The contribution of the mean particle diameter of the erodent is predominant compared to the substrate conditions.

**Keywords:** high-chromium cast irons; solid particle erosion; heat treatments; microstructure





## 1. Introduction

In industrial applications, a widely adopted solution to extend the service life of components in machinery equipment or implants involves the application of protective coatings to the most critical zones exposed to severe wear conditions (e.g., high operating temperatures, corrosive environments, mechanical damage by impact of solid particles) through welding techniques [1]. Among various surface coating and protective hardening techniques, hardfacing stands out due to its low cost and ease of handling [2–4]. Hardfacing enhances the corrosive, abrasive, and heat resistance properties of the surface by creating a cladding metal layer with improved features.

Typical engineering materials that have proven successful in dealing with wear damages include steels, cast irons, Cu alloys, Co alloys, Ni alloys, and ceramics [5,6]. Cast irons stand out due to their excellent wear resistance, ready availability, and cost-effectiveness High-Chromium Cast Irons (HCCIs), are extensively used for their superior wear resistance in both severe abrasive wear conditions (e.g., grinding media) and erosive (e.g., slurry, gravel, and dredge pumps) applications. Unforeseen failures and subsequent economic losses in such applications are often linked to wear damages. Therefore, there is an ongoing demand for materials with outstanding abrasion wear and corrosion resistance, as well as good heavy load transfer application behavior.

HCCIs derive their properties from a chemical composition typically ranging from 2 to 7 wt.% C and 5 to 36 wt.% Cr [3]. HCCIs can be considered composite materials because their microstructure contains kinds of carbides such as $M_{23}C_6$, $M_7C_3$, and $M_3C_2$ within a softer iron matrix that accounts for their outstanding performance (high hardness and wear resistance). Considering that hardfacing is typically deposited on the substrate through welding techniques, the microstructure of Fe–Cr–C hardfacing alloys resulting from a non-equilibrium solidification process comprises a Fe–Cr solid solution phase and complex carbides, dependent on the Cr and C contents of the alloys [7]. Hypereutectic Fe–Cr–C alloys are typical wear-resistant materials that consist of primary $M_7C_3$ (M = Cr, Fe) carbides with a Vickers hardness ranging from 1300 to 1800, surrounded by a softer matrix of eutectic $M_7C_3$ carbides and austenite. As a result, they exhibit high hardness and superior wear resistance compared to hypoeutectic ones.

Much research has been focused on the performance of these alloys, their microstructural [8,9] and crystallographic features [10–12], and the close relationship between them [13,14]. The superior resistance of hypereutectic alloys arises from the dispersion of hard $M_7C_3$ carbides [15–18]. Despite this, the as-cast condition of hypereutectic HCCIs may not meet the demands of heavy impact conditions due to the difference in hardness between the matrix and the carbides. Hence, while hypoeutectic HCCIs are still employed in high-demanding environments, efforts are directed toward enhancing the wear resistance of both hypo- and hyper-eutectic alloys, achieving an optimal combination of matrix and carbides' hardness and toughness, which is crucial to enhancing overall resistance [19,20]. As a result, heat treatments are performed to promote microstructural changes and, in turn, improve the mechanical properties [21–24]. Compared to adding expensive alloying elements to modify the morphology, size, and distribution of carbides [25,26], this solution is cost-effective in industrial applications where wear resistance is a key objective. As for wear resistance, to date, the majority of studies have examined the characteristics of erodent particles (such as shape, size, velocity, angle of rotation, and impact angle) that influence erosion [20]. However, to the best of the authors' knowledge, the effects of heat treatment and the dimension of the erodent powder have not been investigated yet.

In this scenario, this study is intended to investigate the combined effects of substrate microstructure and erodent particle characteristics on the erosion resistance of the adopted HCCIs. The proposed analysis originates from a specific application of HCCIs, namely the critical surfaces of large industrial fans [27], where conditions of erosive wear from solid particles are a limiting factor for the components' lifespan. The evaluation of the cast iron's erosive wear resistance was carried out in accordance with ASTM G76 standards, using a dedicated test bench and employing commercially controlled granulometry powder as erodent particles. Experimental tests comprised the erosion resistance of the as-received material, the heat-treated material subjected to either a tempering (500 °C for 120 min followed by air cooling) or a destabilization (980 °C for 90 min, followed by air cooling). Furthermore, different mean diameters of the erodent powder were analyzed. Through the measurement of weight loss and the analysis of the morphological characteristics of the worn surface, it was possible to highlight the features of the erosion mechanisms for the considered test combinations. The results revealed that the erosion process is affected by the erodent's mean particle diameter rather than the substrate's condition.

## 2. Materials and Methods

The material employed in this investigation is a commercial hardfacing alloy comprising a high-chromium cast iron overlay deposited by the open-arc welding of a flux-cored wire onto a low-carbon steel substrate. The chemical composition of the HCCI was determined by the Glow Discharge Optical Emission Spectroscopy (GDOES, Spectruma Analytik GDS 650, Hof, Germany) technique and is detailed in Table 1.

**Table 1.** Chemical composition (wt.%) of the analyzed HCCI.

| Composition (wt.%)—Fe Balance | | | | | | | |
|---|---|---|---|---|---|---|---|
| **C** | **Mn** | **Si** | **Cr** | **Mo** | **Nb** | **W** | **V** |
| 4.15 | 0.56 | 1.08 | 21.04 | 2.78 | 4.09 | 0.86 | 0.69 |

From the commercially available HCCI plate, samples with dimensions of 20 mm × 20 mm × 10 mm were obtained by water cutting. These dimensions were chosen in accordance with the characteristics of the test bench used for erosion tests, described below.

The first step comprised the metallographic characterization of the material in the as-received condition using a Leica DMi8 A light optical microscope (LOM) (Leica, Wetzlar, Germany) and a Zeiss EVO MA15 (Carl Zeiss, Jena, Germany) scanning electron microscope (SEM), equipped with an Oxford X-Max 50 (Oxford Instruments, Abingdon-on-Thames, UK) Energy Dispersive Spectroscopy (EDS) X-ray microprobe for semi-quantitative chemical analysis. SEM micrographs were captured using a secondary electron detector (SEI-SEM) and a backscattered electron detector (BSE-SEM). Microstructural analysis was conducted after standard metallographic preparation (up to 3 μm diamond polishing) and chemical etching with Kalling's reagent (5 g $CuCl_2$, 100 mL HCl, 100 mL $C_2H_5OH$) for 5 s to identify the metallographic constituents. Macro Vickers hardness was also measured by an automatic QATM Qness 60 CHD Master + (QATM, Golling an der Salzach, Austria) hardness tester, according to the UNI EN ISO 6507:2018 standard [28]. The mean Vickers macro-hardness, using an applied load of 30 kgf (HV30) was calculated from five indentations to check the reproducibility of the data.

To investigate the role of the particle size of the erodent powder on erosion resistance, an analysis of the microstructural features of the HCCI was developed. By using the ImageJ image analysis software (Version 1.53e, 2020, National Institutes of Health, Bethesda, MD, USA) [29], the relative distances (named edge-to-edge distance) between primary $M_7C_3$ carbides were evaluated. Image analysis was conducted on each of the 9 samples used for the erosion tests: the surface was divided into 6 regions of interest (ROI), and the edge-to-edge distances were obtained for each ROI, covering a total area of 9 mm² per sample. More in detail, from the raw optical micrograph, the edge-to-edge (E-E) distances between carbide and its adjacent ones were evaluated by the image post-processing sequence displayed in Figure 1. From the original optical micrograph, the color threshold value for the identification of the features under examination was set, the numerical threshold for the evaluation of only the primary $M_7C_3$ carbides was set, and finally, the post-processed binary image to obtain the matrix distances of each carbide and its adjacent ones was obtained. To analyze the relative distances, a tailored Matlab® code was employed. Based on these findings, the sizes of the erodent powder were determined by considering two possible conditions: (i) the edge-to-edge distance greater than the mean diameter of the erodent powder, and (ii) the edge-to-edge distance less than the mean diameter of the erodent powder. Note that, from the same image analysis procedure, the area of the carbides was also evaluated and used to determine the so-called carbide volume fraction (CVF).

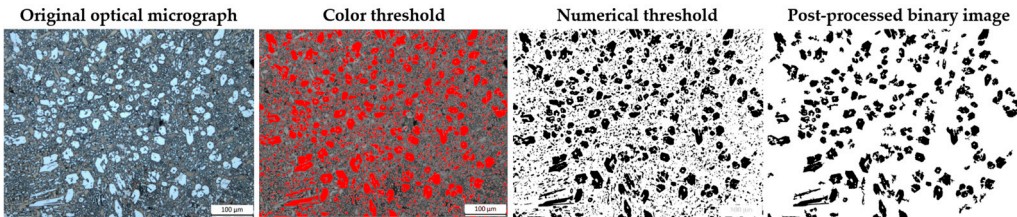

**Figure 1.** Image post-processing sequence to evaluate the edge-to-edge distance between carbides: original optical micrograph, color threshold value setting, numerical threshold setting, and post-processed binary image.

Then, two heat treatment routes were performed to assess the effect of heat treatment on the microstructural, hardness, and erosion resistance characteristics of the material. The first one (named HT1) comprised a tempering treatment conducted at 500 °C for 120 min, followed by air cooling to promote the relaxation of residual stresses resulting from the welding production process. Furthermore, a destabilization heat treatment (named HT2) was conducted at 980 °C for 90 min, followed by air cooling to induce the destabilization of austenite and the precipitation of secondary carbides in the matrix. All the aforementioned treatments were conducted in an LTF (Lenton Furnaces and Ovens, Hope, UK) tube furnace. The subsequent characterization involved optical and scanning electron microscopy techniques, along with hardness and microhardness tests, following the methodologies described earlier.

Erosive wear tests were conducted through a test bench designed following the specifications outlined in the ASTM G76 standard [30]. The adopted test bench provided flexibility in adjusting parameters such as air flow rate, injected powder flow rate, impact angle, and test duration. Additional information on its operational conditions is available in [21,31]. To replicate the most severe and frequent damaging conditions in which such hardfacing operates under operational conditions [27], an impact angle of 15° was adopted. Test parameters, i.e., flow rate, injected powder flow rate, and test duration, were determined through computational fluid dynamics simulation, maintaining a working condition with constant kinetic energy of erodent particles. As for the erodent powder, the controlled-grain-size standard Arizona dust quartz (ARIZ-ISO) commercial powder was chosen. This standard powder (ISO 12103-1:2016 standard [32]) is composed of silica dioxide (75%), aluminum trioxide (20%), and other minor oxides (magnesium, iron, etc.), and it is commonly used for testing filtration systems and machine degradation [33]. The erosion resistance was evaluated by the sample weight loss, measured using a Kern ABT 100-5NM (Kern, Balingen, Germany) analytical balance with an accuracy resolution of 0.01. The so-called erosion rate (ER) parameter, which is the ratio of the weight lost by the sample to the mass of injected erodent, was related to the different substrate-erodent combinations. The ER was averaged three times for each combination. On the worn surfaces, the SEM investigations enabled the detection of the wear mechanisms.

## 3. Results and Discussion

Figure 2 displays optical micrographs of the investigated HCCI in the as-received condition. The microstructure comprised pro-eutectic carbides of the $M_7C_3$ type, both in hexagonal and elongated forms, appearing as light polygons. The observed morphology, i.e., an irregular polygonal shape with hollows at the center and gaps on the edge, has been widely analyzed in numerous studies [10,34]. The disclosed morphology resulted from solidification conditions occurring during the welding deposition of the hardfacing onto the steel substrate, as observed by Chung et al. [15]. Compacted NbC carbides, integrated with Cr-rich carbides, can be observed in the microstructure (Figure 2b). This form has been related to the growing process: NbC carbides may have acted as nucleators of $M_7C_3$ primary carbides, similarly to the nucleation of austenite dendrites [16,35].

The quantitative image analysis of the observed microstructure revealed a mean CVF of $24 \pm 3\%$. Several authors propose to relate the CVF parameter with the erosion resistance of the material [17,18,36]. However, this parameter does not account for the shape and distribution of the carbides but only indicates the fraction of the area occupied by them. Hence, to obtain an effective parameter addressing the role of microstructure and erodent powder characteristics, the edge-to-edge distances of adjacent $M_7C_3$ primary carbides were evaluated. From the image analysis, the mean E-E distances resulted in $12 \pm 2$ μm, $9 \pm 3$ μm, and $11 \pm 2$ μm for the as-received HT1 and HT2 conditions. This experimental observation aligns with earlier research [14], indicating that eutectic carbides remain largely unaffected by heat treatments, while the treatments primarily induce microstructural modifications in the matrix, as detailed below.

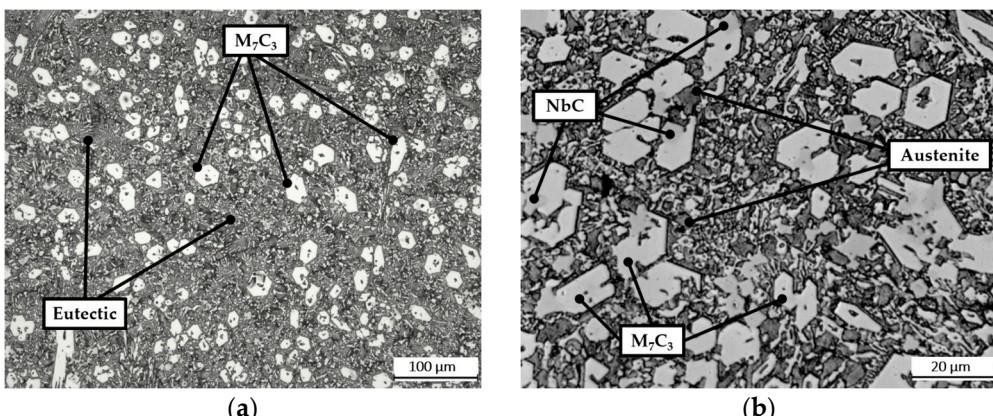

**Figure 2.** Optical micrographs of the investigated HCCI in the as-received condition: (**a**) pro-eutectic $M_7C_3$ carbides together with the eutectic constituent; (**b**) $M_7C_3$ and NbC carbides, austenite.

SEM analyses of the HCCIs were conducted on as-received and heat-treated conditions to detect the microstructural changes that occurred (Figure 3). As for the as-received condition (Figure 3a), the BSE-SEM micrograph revealed a microstructure consisting of pro-eutectic $M_7C_3$ carbides surrounded by a matrix of eutectic carbides, austenite, small amounts of martensite, Nb-rich carbides, and Mo-rich phases. These microstructural constituents were observed and characterized in a previous study by the Authors [21], where the coupled SEM and X-ray diffraction analyses enabled the identification of the austenitic microstructure and traces of martensite at the matrix/carbide interface, as the result of the local depletion of the austenite from alloying addition, as reported by Tabrett and Sare [37]. After the HT1 tempering treatment, as reported by previous studies, a reduction of the amount of retained austenite in the matrix and residual stress occurs [22,34]. Moreover, it has been recently pointed out that the primary and eutectic carbides do not change significantly with tempering parameters [38]. After tempering, see the BSE-SEM micrograph of Figure 3b, together with the corresponding EDS spectra of Figure 3d, the microstructure revealed the same microstructural constituents. Finally, the HT2 destabilization treatment (Figure 3c and the corresponding EDS spectra of Figure 3e) promoted the overall transformation of austenite into martensite, with the dissolution of eutectic carbides and the precipitation of secondary carbides. In a previous study by the Authors [21], it was detected through X-ray diffraction analyses that such destabilization treatment led to an increase in the martensite content (from 22.3 wt.% for the as-received condition to 56.9 wt.% after the destabilization), which was related to the reduction of the retained austenite content that was reduced from 33.3 wt.% for the as-received condition to 1.1 wt.% after the destabilization.

The influence of the heat treatments was explored through Vickers hardness testing. The bulk values resulted in $621 \pm 9$ HV30 for the as-received condition, $586 \pm 3$ HV30 for the HT1 treatment, and $968 \pm 11$ HV30 for the HT2 treatment. As expected, HT1 is effective in reducing the overall bulk hardness and HT2 is effective in increasing the bulk hardness due to the transformation of austenite to martensite of the matrix [21,23,24,39].

Based on the results of the image analysis, two mean particle diameters of the standard Arizona dust quartz powder were chosen: dmean = 4.8 μm and dmean = 25.5 μm. The powder characterization comes from a specific size characterization using the Mastersizer 3000 laser diffraction analyzer (Malvern Panalytical, Malvern, UK). The powder with smaller mean particle sizes was labeled as UF, while the larger one was labeled as M. The BSE-SEM images of the adopted powders are presented in Figure 4, where the irregular grain shape can be detected.

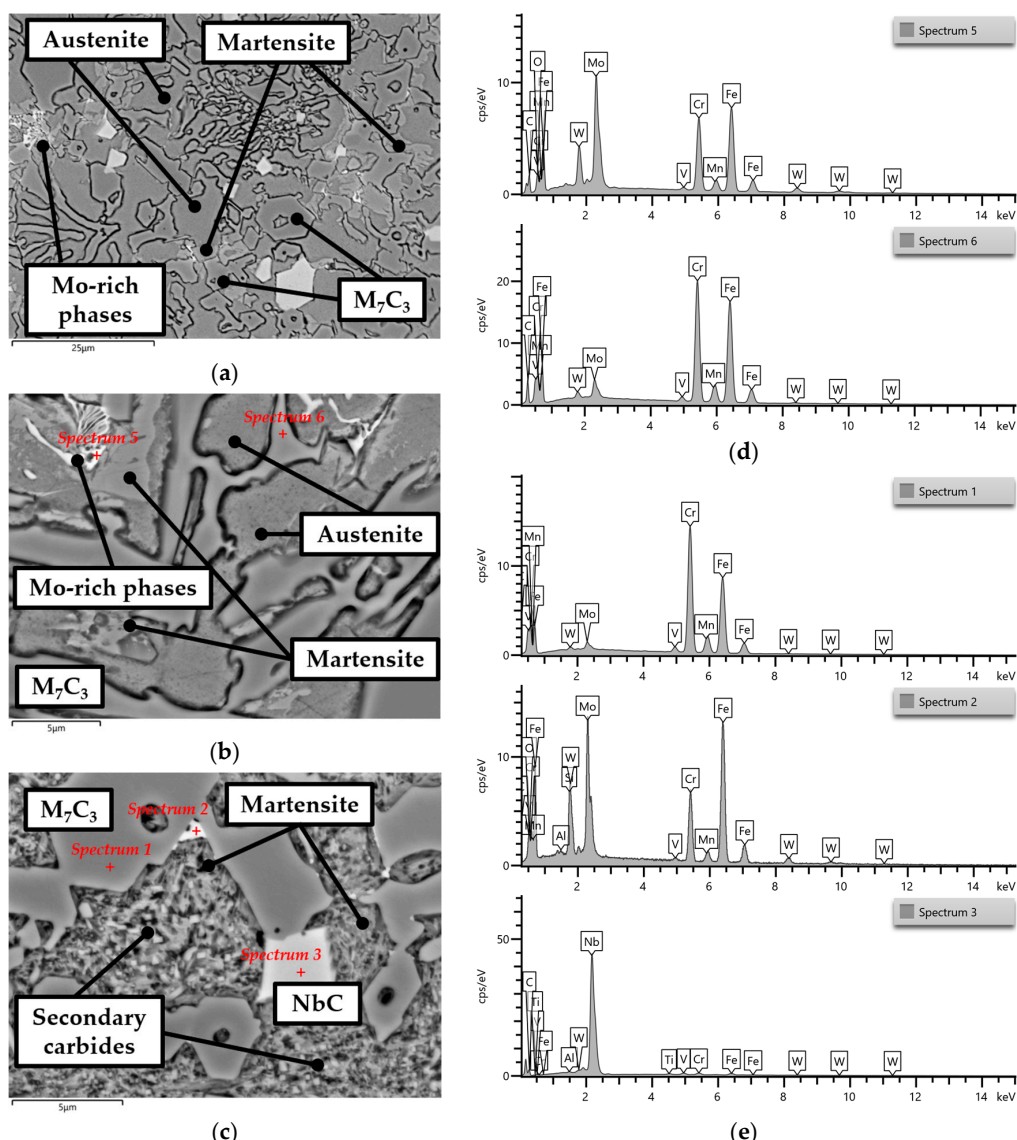

**Figure 3.** BSE-SEM micrographs (**a–c**) and EDS spectra (**d,e**) of the investigated HCCIs in: (**a**) as-received, (**b**) HT1, and (**c**) HT2 conditions. (**d**) EDS spectra for the point analyses in (**b**); (**e**) EDS spectra for the point analyses in (**c**).

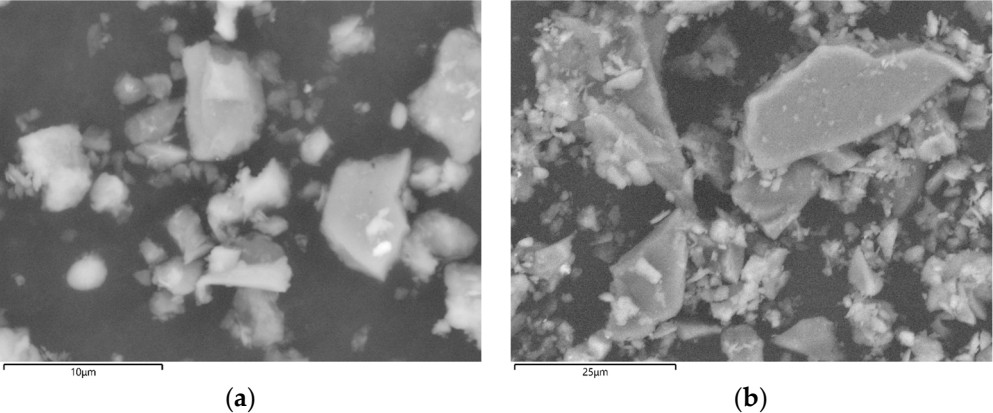

**Figure 4.** BSE-SEM images of the Arizona dust quartz bunch powder used in erosion tests: (**a**) UF powder with a dmean = 4.8 μm and (**b**) M powder with a dmean = 25.5 μm.

As for the erosion tests, test parameters were computed for the UF and M powders by computational fluid dynamic simulations. The mass-averaged impact velocity was assessed by numerical simulations with reference to the powder diameter distribution. The numerical strategy and the model have already been used in [21,40] to assess the flow field in front of the target (i.e., stagnation region) and the related particle impact behavior. The obtained results are summarized in Table 2.

**Table 2.** Parameters of the erosion tests computed by computational fluid dynamic simulations.

| Test Parameters | UF Powder (dmean = 4.8 μm) | M Powder (dmean = 25.5 μm) |
|---|---|---|
| Air mass flow [kg/s] | 0.0036 | 0.0003 |
| Flow rate [L/min] | 186 | 8.2 |
| Particle impact velocity [m/s] | 205 | 15.9 |

It has been reported that the ER increases with the increase in particle impact velocity, and a power law relationship has been used to correlate the ER and particle impact velocity [20]. The particle velocity is the key variable in the modeling process of the erosive issues. At the same time, erosion resistance is a challenging phenomenon related, among others, to particles' features (e.g., size, shape, hardness, and impact angle), targets' features (e.g., material, hardness, and chemical composition), and boundary conditions (e.g., temperature and carrier fluid) [20]. Due to this, since the substrate characteristics change in the present investigation, to cross-correlate the results, the tests are carried out by keeping the amount of particle energy comparable. Therefore, smaller particles have greater velocity instead of bigger ones.

The combined effects of substrate microstructure and the erodent particle characteristics on the erosion resistance of the investigated HCCIs are summarized in Figure 5, which displays the ER as a function of the mean particle diameters of the erodent powder for the investigated conditions, namely AR, HT1, and HT2. Note that the uncertainty band associated with the ER measurements has the same size as the markers on the charts. As observed, regardless of the substrate conditions, the ER decreases with increasing the mean particle diameters of the erodent powder, indicating that the severity of the erosive phenomenon is greater for particles with smaller mean particle diameters. This result is in line with previous studies that investigated the dependence of particle impact velocity on erosion resistance and detailed the factors affecting the velocity exponent value [41–43]. In this condition, the size of the particle drives the erosion phenomenon. Looking at the effects of the heat treatment for the HT1 and HT2 conditions, a reduction of the particle velocity effects is recognizable. The differences between high-velocity and low-velocity conditions were reduced in the heat-treated samples with respect to the as-received ones. The modification of the matrix characteristics due to the heat treatment modified the overall erosion resistance characteristics of the substrate, as detailed below.

The SEM analyses of the worn surfaces for the AR, HT1, and HT2 conditions are summarized in Figures 6 and 7 for the UF and M powders, respectively. The low-magnification SEM micrographs (Figures 6a–c and 7a–c) report the SEI-SEM image on the left and the BSE-SEM image on the right. By comparing the worn surfaces of the HCCIs eroded by UF powder (dmean = 4.8 μm), see Figure 6, with the ones eroded by M powder (dmean = 25.5 μm), see Figure 7, the contribution of the mean diameter of the erodent appeared predominant to the substrate conditions. Remarkable and wide surface damage can be detected (see Figure 6), and the particles' flow direction (from the top to the bottom of each image) can be identified (see the SEI-SEM images of Figure 6a–c).

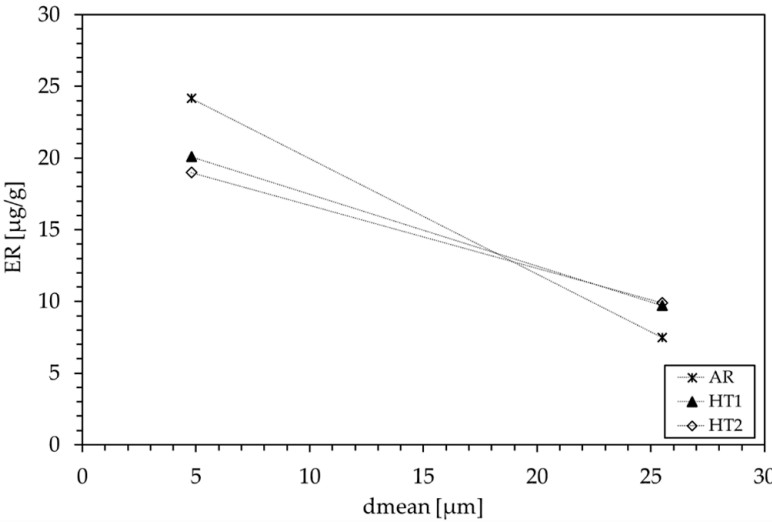

**Figure 5.** Erosion rate (ER) against the mean particle diameters of the erodent powder (dmean) for the as-received (AR), heat-treated at 500 °C for 120 min (HT1), and heat-treated at 980 °C for 90 min (HT2) conditions. The uncertainty band associated with the ER is the same size as the markers on the charts.

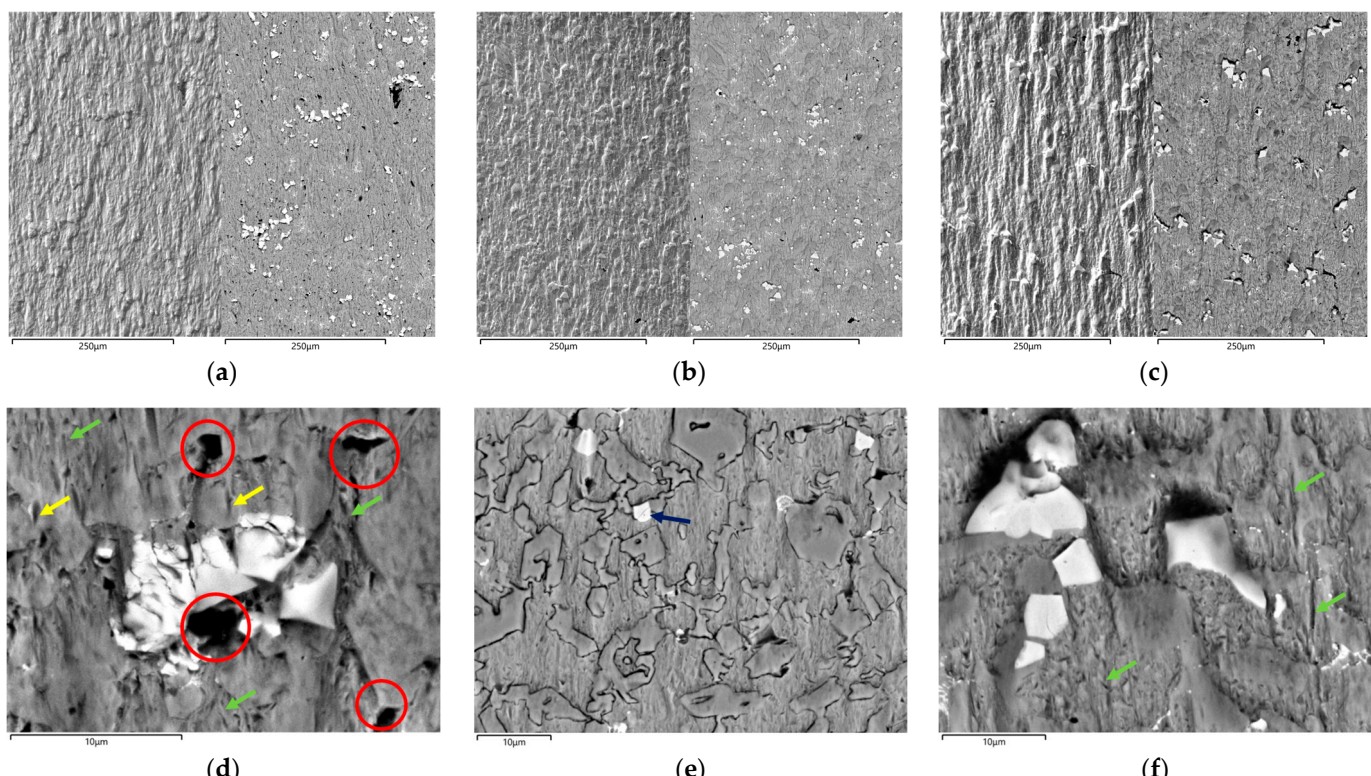

**Figure 6.** SEM micrographs of the worn surfaces for the AR (**a**,**d**), HT1 (**b**,**e**), and HT2 (**c**,**f**) conditions. Surfaces eroded by powder with a dmean = 4.8 μm. (**a**–**c**) report the SEI-SEM image on the left and the BSE-SEM image on the right; (**d**–**f**) are blow-ups of (**a**–**c**), respectively.

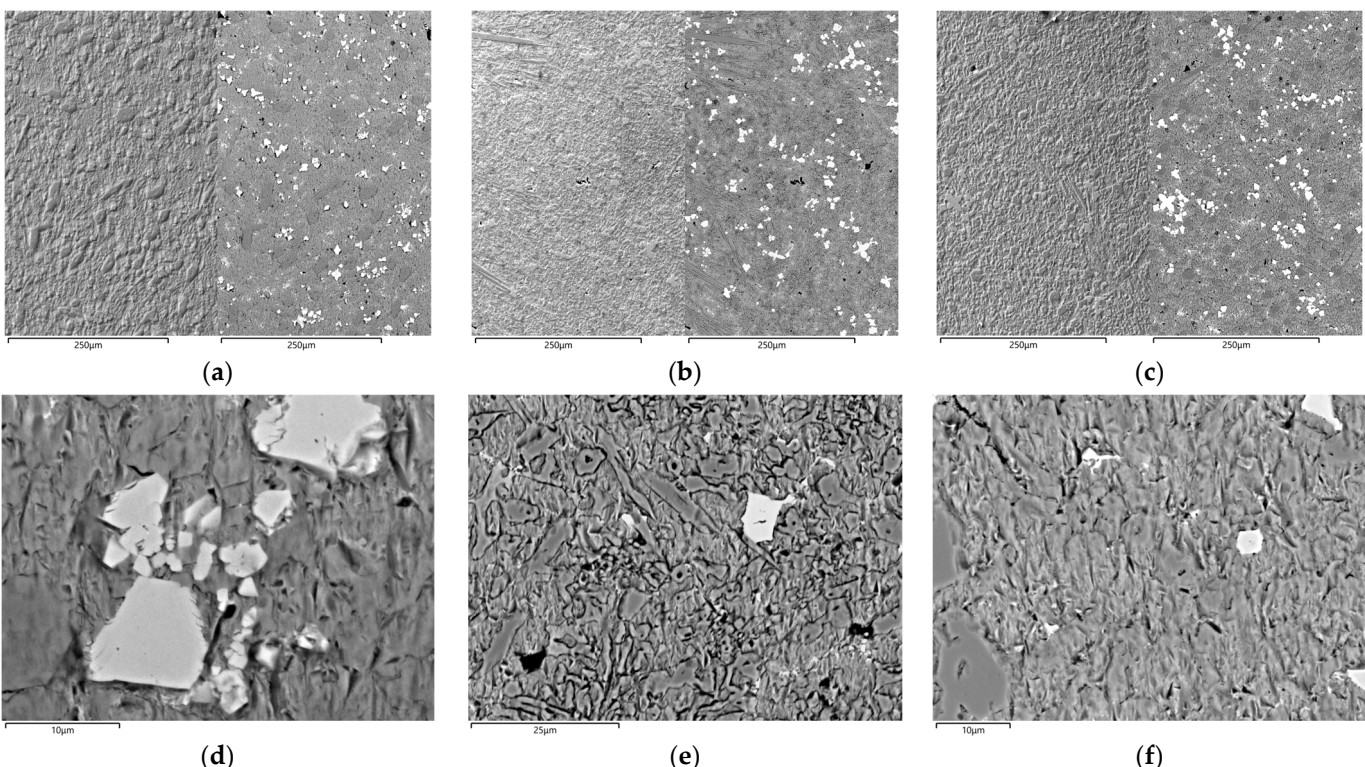

**Figure 7.** SEM micrographs of the worn surfaces for the AR (**a**,**d**), HT1 (**b**,**e**), and HT2 (**c**,**f**) conditions. Surfaces eroded by powder with a dmean = 25.5 μm. (**a**–**c**) report the SEI-SEM image on the left and the BSE-SEM image on the right; (**d**–**f**) are blow-ups of (**a**–**c**), respectively.

From the blow-ups (see Figure 6d–f), the role of the microstructure was deepened. In the as-received condition (Figure 6d), the Nb-rich carbides appeared fractured and cracked due to their high hardness and brittleness. In addition, craters (red circles) and micro-cutting (green arrows) damage were detected on the matrix; plow scars were detected on the eutectic carbides (yellow arrows). In the HT1 heat-treated condition (see Figure 6e), the Nb-rich carbides appeared fractured (dark blue arrow), although the eutectic carbides were less damaged in comparison with the as-received condition. It is likely that the tempering treatment, promoting the relaxation of residual stresses, reduced the overall brittleness of the microstructure. On the contrary, the HT2 condition displayed wide surface damage: deep and straight plow scars (green arrows) were visible. As seen, the Nb-rich carbides, which act as a barrier to plowing, were fractured, and due to the high hardness of the matrix, they tended to fall off, as already observed in a previous study [44]. Such damage was further detailed by the EDS elemental maps in Figure 8, which displays the worn surfaces for the HT2 condition eroded by UF powder. As reported elsewhere [45], for the heat-treated conditions, the plastic deformation of the matrix is less pronounced; in addition, the primary and secondary carbides enhance the erosion resistance of the alloys. Harder Nb-rich carbides, which act as protection from erosion damage, resulted in cracking and being pulled out of the matrix, as suggested by the altered morphologies on the worn surface (Figure 8).

Concerning the analysis of the worn surfaces of the HCCIs eroded by M powder (Figure 7), the particles' flow direction was not detectable, and all the surfaces appeared uniformly damaged. The BSE-SEM observations suggest that the worn surfaces eroded by M powder, see Figure 7a–c, contain a higher amount of NbC carbides with respect to the worn surfaces eroded by UF powder, see Figure 6a–c. From the blow-ups of Figure 7d–f, the role of Cr-rich and Nb-rich carbides was deepened: due to the higher erodent's particle dimension, the impact on the substrate was faced by a greater number of carbides, and thus, each of these was subjected by a lower amount of energy. From the EDS elemental

maps of the worn surfaces of the HT1 condition eroded by UF powder (Figure 9), the contribution of the mean particle diameter is predominant to the microstructural changes, as also confirmed in the above-reported ER outcomes.

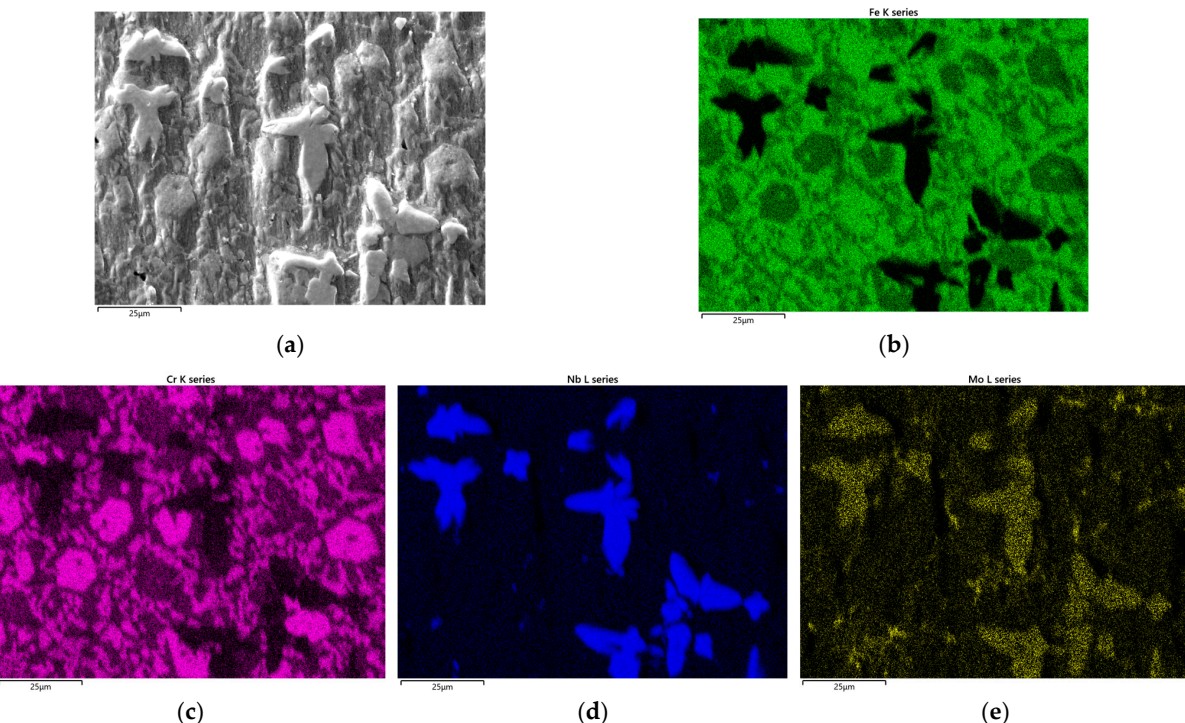

**Figure 8.** (**a**) SEI-SEM micrograph of the worn surfaces for the HT2 condition. Surfaces eroded by powder with a dmean = 4.8 μm. (**b**–**e**) EDS maps of the elemental distribution of iron, chromium, niobium, and molybdenum, respectively.

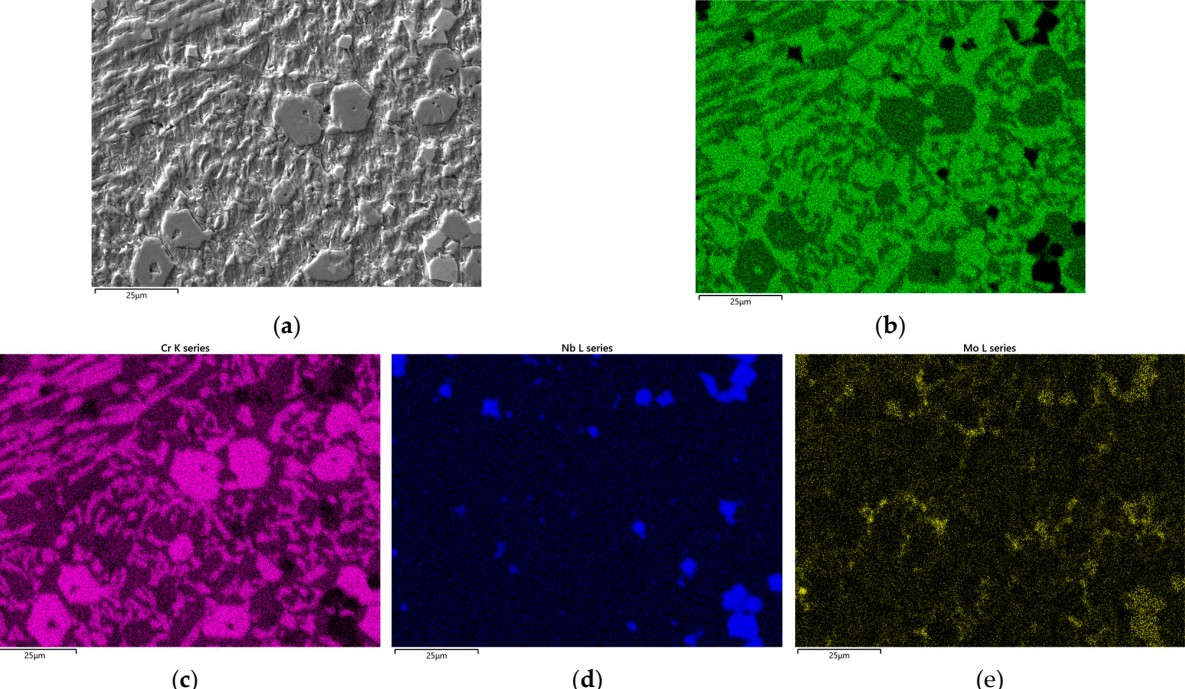

**Figure 9.** (**a**) SEI-SEM micrograph of the worn surfaces for the HT1 condition. Surfaces eroded by powder with a dmean = 25.5 μm. (**b**–**e**) EDS maps of the elemental distribution of iron, chromium, niobium, and molybdenum, respectively.

## 4. Conclusions

The present study seeks to investigate the solid particle erosion resistance of an HCCI, focusing on the combined role of substrate conditions and erodent powder characteristics. From the experimental findings, the following conclusions can be drawn:

- the proposed heat treatments, i.e., tempering and destabilization, changed the hardness and the microstructure of the HCCI; nevertheless, the effects on the erosion resistance were minor compared to the contribution of the erodent particle size;
- the ER values decreased with increasing the mean diameter of the erodent, indicating that the severity of the erosive phenomenon is heavier for the smaller erodent's mean particle diameter;
- from the analysis of the worn surfaces, the role of the microstructure in the erosion mechanisms was deepened: Nb-rich carbides act as a barrier to plowing. When the mean diameter of the erodent is lower than the edge-to-edge distances between the carbide and its adjacent ones, they were fractured and tended to fall off. In addition, the plastic deformation of the matrix is remarkable. Conversely, for a higher mean diameter of the erodent, the impact on the substrate was faced by a greater number of carbides, resulting in lower ER values;
- the experimental findings suggested that the contribution of the mean diameter of the erodent is predominant compared to the substrate conditions.

This study provides insights into the combined effects of heat treatment and erodent particle size on the erosion behavior of high-chromium cast iron. The results show that the erosion-resistant characteristics of a specific substrate should include the powder characteristics in terms of dimension and velocity and, in the case of carbide-reinforced materials, the particle dimensions are crucial. Further investigation will be devoted to cross-correlating these data to generalize this erosive phenomenon. The findings contribute to understanding the interaction between substrate microstructure, treatment conditions, and erodent characteristics in solid particle erosion.

**Author Contributions:** Conceptualization, A.F. and A.S.; methodology, A.S.; software, A.S.; formal analysis, A.F.; investigation, A.F. and A.S.; resources, A.S.; data curation, A.F.; writing—original draft preparation, A.F.; writing—review and editing, A.S.; visualization, A.F. All authors have read and agreed to the published version of the manuscript.

**Funding:** This research received no external funding.

**Institutional Review Board Statement:** Not applicable.

**Informed Consent Statement:** Not applicable.

**Data Availability Statement:** The data presented in this study are available in the article.

**Acknowledgments:** The authors wish to gratefully acknowledge Andrea Cordone for his support during the experimental campaign. Our gratitude is also extended to Nicola Zanini for his valuable contribution to the erosion tests.

**Conflicts of Interest:** The authors declare no conflict of interest.

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
