# Peer review of "Effects of Heat Treatment and Erosion Particle Size on Erosion Resistance of a Hypereutectic High-Chromium Cast Iron"

_coatings, doi:10.3390/coatings14010066_

Round 1
Reviewer 1 Report
Comments and Suggestions for Authors
The author needs to incorporate the below suggestions/comments in their manuscript so that the reader's clarity should be clearer.
1. The title of the manuscript needs to be clearer as per the work because the author has used hypereutectic high chromium cast iron.
2. The abstract must include at least one or two lines about the work's need and application. Further, it should be more quantitative in terms of improvement.
3. The author should add a few more papers from 2023 so that the novelty of the work is improved.
4. The author should explore more Erosive wear tests in terms of test conditions.
5. The author should present the XRD analysis to clarify phases present in pure and heat-treated alloys.
6. The author must justify the comparison of their study on two different parameters for different particle sizes.
7. The worn-out surface also required the XRD analysis and also author presented the findings inside the SEM images. Area mapping will also be helpful for the same.
8. The author should include the parameters of Vikers hardness test.
9. The author must include all the results in the conclusion.
Author Response
ANSWERS TO REVIEWERS
The Authors wish to thank the Reviewers for their valuable revision and for all the insightful suggestions and corrections reported.
Based on the Reviewers' suggestions, the Authors have clarified many parts of the work. Detailed considerations have been incorporated into the manuscript (highlighted in yellow in the text) to fulfill the Reviewers' recommendations and improve the overall quality of the paper. The Authors have reported their considerations and answers to each point raised by every Reviewer.
REVIEWER #1
The author needs to incorporate the below suggestions/comments in their manuscript so that the reader's clarity should be clearer.
- The title of the manuscript needs to be clearer as per the work because the author has used hypereutectic high chromium cast iron.
The Authors would like to thank the Reviewer for his/her valuable comment.
As suggested, the title has been changed to read "Effects of heat treatment and erosion particle size on erosion resistance of a hypereutectic high chromium cast iron."
- The Abstract must include at least one or two lines about the work's need and application. Further, it should be more quantitative in terms of improvement.
The Authors would like to thank the Reviewer for his/her valuable comment.
As indicated, additional quantitative details have been included to highlight the specific improvements achieved through the research to enhance the Abstract's clarity.
The Abstract has been amended as follows:
" This research addresses the erosive resistance of a hypereutectic high chromium cast iron subjected to solid particle erosion. The study stems from a specific application of high chromium cast iron, i.e. the critical surfaces of large industrial fans operating in a cement clinker grinding plant where such damage is a limiting factor for the components' lifespan. A dedicated experimental investigation on the impact of substrate microstructure and erodent particle size on erosion resistance was set. The experimental campaign, conducted in a dedicated test bench per the ASTM G76 standard, comprised the analysis of the as-received, tempered and destabilized conditions for the cast iron. From a preliminary image analysis of the microstructural features, two diameters of the erodent powder for the erosion tests were defined. The observed erosion rate decreased with the increase in the mean particle diameter of the erodent, indicating more severe erosive conditions for smaller particles. From the analysis of the worn surfaces, it was possible to highlight the involved mechanism concerning the considered test combinations. For the as-received condition, the erosion rate with the larger mean particle diameter of the erodent decreased three times compared to the smaller one. For the heat-treated conditions, the erosion rate was halved with the larger mean particle diameter of the erodent. The proposed analysis, intended to get more insight into the limiting factor for the components' lifespan for erosive wear damage, proved that erosion resistance is not dependent on the material hardness. The contribution of the mean particle diameter of the erodent is predominant compared to the substrate conditions."
- The author should add a few more papers from 2023 so that the novelty of the work is improved.
The Authors would like to thank the Reviewer for his/her comment.
The following papers have been added:
Yawei, L.; Wei, L.; Penghui, Y.; Hanguang, F.; Wenhang, Y.; Tounan, J.; Zhengyang, C. Microstructure Evolution and Hardness of Hypereutectic High Chromium Cast Iron after Tempering. J. Mater. Eng. Perform. 2023.
Gaqi, Y.; Kusumoto, K.; Shimizu, K.; Purba, R.H. Effect of Carbide Orientation on Wear Characteristics of High-Alloy Wear-Resistant Cast Irons. Lubricants 2023.
Rajicic, B.M.; Maslarevic, A.; Bakic, G.M.; Maksimovic, V.; Djukic, M.B. Erosion Wear Behavior of High Chromium Cast Irons. Trans. Indian Inst. Met. 2023, 76, 1427–1437.
- The author should explore more Erosive wear tests in terms of test conditions.
The Authors would like to thank the Reviewer for his/her insightful suggestion.
The primary aim of our study was indeed to replicate the most severe conditions in which such hardfacing operates, as highlighted in a previous study by the Authors (https://doi.org/10.1016/j.engfailanal.2023.107058), with a specific focus on understanding the role of the mean particle diameter of the erodent powder rather than explore the most common factors affecting the surface erosion resistance. The Authors are aware that, among others, particle shape, particle impact angle, relative hardness of erodent particles, and substrate play a crucial role. In this regard, the Authors deepened the effect of the mean particle diameter of the erodent and, based on the obtained results (i.e. the contribution of the mean particle diameter of the erodent is predominant compared to the substrate conditions), are going to explore the combined effect of the mean particle diameter of the erodent and impact angles. The latter is still an open point in the literature; although extensive research has been conducted, there is no general agreement about the behavior (ductile and/or brittle) and dependence on impact angle for high chromium cast irons subjected to solid particle erosion. The ongoing studies will contribute to this and the robustness of the present findings.
- The author should present the XRD analysis to clarify phases present in pure and heat-treated alloys.
The Authors would like to thank the Reviewer for his/her constructive comment.
The Authors have amended the description of the metallographic constituent of the alloy in the as-received and heat-treated conditions. In addition, the Authors have detailed the analysis through SEM and EDS investigations: Fig. 3 has been changed and the corresponding description has been rewritten based on relevant scientific literature.
- The author must justify the comparison of their study on two different parameters for different particle sizes.
The Authors would like to thank the Reviewer for his/her valuable comment.
Based on data from the literature, focusing on the role of particle impact velocity on erosion resistance and effectively taking into account the role of the substrate microstructures in affecting the erosion resistance, the Authors set the experimental tests to have the kinetic energies for the two adopted mean particle diameters at a similar order of magnitude. Such adopted test conditions have been detailed in the manuscript.
- The worn-out surface also required the XRD analysis and also author presented the findings inside the SEM images. Area mapping will also be helpful for the same.
The Authors would like to thank the Reviewer for his/her helpful comment.
Regrettably, the Authors are not able to provide XRD analysis on the worn-out surfaces due to the unavailability of the XRD equipment. As suggested, the EDS area mapping has been added to the results.
- The author should include the parameters of Vikers hardness test.
The Authors would like to thank the Reviewer for his/her suggestion.
The parameters of the hardness tests have been detailed in the Materials and Methods section.
- The author must include all the results in the conclusion.
The Authors would like to thank the Reviewer for his/her valuable comment. As suggested, the Conclusion has been amended as follows:
" The present study seeks to investigate the solid particle erosion resistance of an HCCI, focusing on the combined role of substrate conditions and erodent powder characteristics. From the experimental findings, the following conclusions can be drawn:
- the proposed heat treatments, i.e. tempering and destabilization, changed the hardness and the microstructure of the HCCI; nevertheless, the effects on the erosion resistance were minor compared to the contribution of the erodent particle size;
- the ER values decreased with increasing the mean diameter of the erodent, indicating that the severity of the erosive phenomenon is heavier for the smaller erodent's mean particle diameter;
- from the analysis of the worn surfaces, the role of the microstructure in the erosion mechanisms was deepened: Nb-rich carbides act as a barrier to plowing. When the mean diameter of the erodent is lower than the edge-to-edge distances between the carbide and its adjacent ones, they were fractured and tended to fall off. In addition, the plastic deformation of the matrix is remarkable. Conversely, for a higher mean diameter of the erodent, the impact on the substrate was faced by a greater number of carbides, resulting in lower ER values;
- the experimental findings suggested that the contribution of the mean diameter of the erodent is predominant compared to the substrate conditions."
Reviewer 2 Report
Comments and Suggestions for Authors
In the present research, the authors try to investigate the effects of heat treatment and particle size on erosion resistance of high chromium cast iron. The authors exhibit some results, but there are some questions.
1. The authors titled the paper as “A comparative analysis of heat treatment and particle size effects on erosion resistance in high chromium cast iron”. It is just a simple research on the influence of heat treatment and erosion particle size on erosion resistance. The title could be simplified as “effects of heat treatment and erosion particle size on erosion resistance of high chromium cast iron”.
2. In the experimental, the authors have claimed that the EDS and X-ray microprobe have been applied to perform the chemical analysis. It wonders where is the chemical composition analysis results in the content. If there is no, they are not necessary to describe in the experimental.
3. In the content, it wonders why there is no no phase characterization. For example, the XRD analysis could be applied to analyze the precipitates.
4. In the content, the authors mark the phases as M7C3 and NbC in the high chromium cast iron. It wonders how the authors determine these phase?
5. In the content, the authors mark some region as Mo rich phase. The authors should provide the corresponding proof to support their opinion.
6. Based on the SEM images of standard Arizona dust quartz powder used in erosion tests. It seems the size of the erosion particles are not equal to the claimed size. Then it wonders how could the authors evaluate the corresponding effect?
7. In the figure 5, it wonders whether there is no error bars? Have not the authors repeated the tests to obtain the reliable data、
8. In the content, the authors have observe the worn surface of the specimens. However, the SEM method could not provide more details on the surface morphology. The laser scanning confocal microscope may be applied to characterize the surface morphology. In addition, what is about the surface chemical composition distribution? That is helpful to understand the effect of microstructure.
9. In the content, the worn surface observations could demonstrate the effects of erosion particle and microstructure. However, it could not exhibit role of defined phases during the erosion and wear processing? The cross-sectional microstructure could be observed to demonstrate the related information. They could refer the previous research “Investigation on microstructure and wear behavior of the NiAl–TiC–Al2O3 composite fabricated by self-propagation high-temperature synthesis with extrusion. Journal of alloys and compounds 2013,554, 182-188”.
Author Response
ANSWERS TO REVIEWERS
The Authors wish to thank the Reviewers for their valuable revision and for all the insightful suggestions and corrections reported.
Based on the Reviewers' suggestions, the Authors have clarified many parts of the work. Detailed considerations have been incorporated into the manuscript (highlighted in yellow in the text) to fulfill the Reviewers' recommendations and improve the overall quality of the paper. The Authors have reported their considerations and answers to each point raised by every Reviewer.
REVIEWER #2
In the present research, the authors try to investigate the effects of heat treatment and particle size on erosion resistance of high chromium cast iron. The authors exhibit some results, but there are some questions.
- The authors titled the paper as "A comparative analysis of heat treatment and particle size effects on erosion resistance in high chromium cast iron". It is just a simple research on the influence of heat treatment and erosion particle size on erosion resistance. The title could be simplified as "effects of heat treatment and erosion particle size on erosion resistance of high chromium cast iron".
The Authors would like to thank the Reviewer for his/her valuable comment. The title has been amended as follows: "Effects of heat treatment and erosion particle size on erosion resistance of a hypereutectic high chromium cast iron".
- In the experimental, the authors have claimed that the EDS and X-ray microprobe have been applied to perform the chemical analysis. It wonders where is the chemical composition analysis results in the content. If there is no, they are not necessary to describe in the experimental.
The Authors would like to thank the Reviewer for pointing this out.
The Authors apologize for this. In the revised version of the manuscript EDS spectra and EDS maps (Figs. 3, 8 and 9) have been added and thus, the EDS equipment has been described in the materials and methods section.
- In the content, it wonders why there is no no phase characterization. For example, the XRD analysis could be applied to analyze the precipitates.
The Authors would like to thank the Reviewer for his/her constructive comment.
The Authors have amended the comments to SEM micrographs of the investigated HCCIs. As for as-received and heat-treated condition, the Authors have provided the phase characterization based on previous analyses. In addition, the Authors have detailed the analysis through SEM and EDS investigations.
- In the content, the authors mark the phases as M7C3 and NbC in the high chromium cast iron. It wonders how the authors determine these phase?
The Authors would like to thank the Reviewer for his/her constructive criticism.
Figure 3 has been amended as suggested by adding the EDS spectra of the relevant phases. The description has been rewritten based on relevant scientific literature.
- In the content, the authors mark some region as Mo rich phase. The authors should provide the corresponding proof to support their opinion.
The Authors would like to thank the Reviewer for his/her constructive criticism.
Figure 3b has been changed by adding the EDS spectrum of the Mo-rich phases.
- Based on the SEM images of standard Arizona dust quartz powder used in erosion tests. It seems the size of the erosion particles are not equal to the claimed size. Then it wonders how could the authors evaluate the corresponding effect?
The Authors would like to thank the Reviewer for pointing this out.
The BSE-SEM images involved only a portion of the powder: this is aimed at showing their irregular grain shape. The characteristic size and distribution cannot be inferred by SEM. Instead, the powder characterization has been evaluated by laser diffraction analysis (Mastersizer 3000 laser diffraction analyzer). All the above-reported information has been added to the manuscript and the caption and the description of Fig. 4 have been amended accordingly.
- In the figure 5, it wonders whether there is no error bars? Have not the authors repeated the tests to obtain the reliable data
The Authors would like to thank the Reviewer for this amendment.
The ER was evaluated by three repeated tests and, for all the investigated conditions, the uncertainty was less than ± 1 µg/g. This has been amended in the materials and methods section. Considering that such a value has the same size as the markers on the chart, the Authors have specified it in the caption and in the corresponding description of Fig. 5.
- In the content, the authors have observe the worn surface of the specimens. However, the SEM method could not provide more details on the surface morphology. The laser scanning confocal microscope may be applied to characterize the surface morphology. In addition, what is about the surface chemical composition distribution? That is helpful to understand the effect of microstructure.
The Authors would like to thank the Reviewer for this helpful suggestion.
The Authors have added a more detailed analysis of the worn surfaces by adding blow-ups of the investigated conditions that enabled a more effective description of the erosion mechanisms. The text has been amended accordingly.
- In the content, the worn surface observations could demonstrate the effects of erosion particle and microstructure. However, it could not exhibit role of defined phases during the erosion and wear processing? The cross-sectional microstructure could be observed to demonstrate the related information. They could refer the previous research "Investigation on microstructure and wear behavior of the NiAl–TiC–Al2O3 composite fabricated by self-propagation high-temperature synthesis with extrusion. Journal of alloys and compounds 2013,554, 182-188".
The Authors would like to thank the Reviewer for this helpful suggestion.
The Authors have added EDS area mapping to corroborate the description of the obtained results.
Reviewer 3 Report
Comments and Suggestions for Authors
In this study, a comparative analysis of heat treatment and particle size effects on erosion resistance in high Cr cast iron was investigated. It is claimed that erosion resistance is independent of material hardness.
The abstract is non-quantitative and missing the novelty. Please rewrite the abstract.
Combine Figure 6-8 and use meaningful results.
The result and discussion portion is very weak, Please rewrite the result and discussion portion and compare the results with the literature.
Author Response
ANSWERS TO REVIEWERS
The Authors wish to thank the Reviewers for their valuable revision and for all the insightful suggestions and corrections reported.
Based on the Reviewers' suggestions, the Authors have clarified many parts of the work. Detailed considerations have been incorporated into the manuscript (highlighted in yellow in the text) to fulfill the Reviewers' recommendations and improve the overall quality of the paper. The Authors have reported their considerations and answers to each point raised by every Reviewer.
REVIEWER #3
In this study, a comparative analysis of heat treatment and particle size effects on erosion resistance in high Cr cast iron was investigated. It is claimed that erosion resistance is independent of material hardness.
The Abstract is non-quantitative and missing the novelty. Please rewrite the Abstract.
The Authors wish to thank the Reviewer for his/her valuable comment.
As suggested, the Abstract has been amended as follows:
"This research addresses the erosive resistance of a hypereutectic high chromium cast iron subjected to solid particle erosion. The study stems from a specific application of high chromium cast iron, i.e. the critical surfaces of large industrial fans operating in a cement clinker grinding plant where such damage is a limiting factor for the components' lifespan. A dedicated experimental investigation on the impact of substrate microstructure and erodent particle size on erosion resistance was set. The experimental campaign, conducted in a dedicated test bench per the ASTM G76 standard, comprised the analysis of the as-received, tempered and destabilized conditions for the cast iron. From a preliminary image analysis of the microstructural features, two diameters of the erodent powder for the erosion tests were defined. The observed erosion rate decreased with the increase in the mean particle diameter of the erodent, indicating more severe erosive conditions for smaller particles. From the analysis of the worn surfaces, it was possible to highlight the involved mechanism concerning the considered test combinations. For the as-received condition, the erosion rate with the larger mean particle diameter of the erodent decreased three times compared to the smaller one. For the heat-treated conditions, the erosion rate was halved with the larger mean particle diameter of the erodent. The proposed analysis, intended to get more insight into the limiting factor for the components' lifespan for erosive wear damage, proved that erosion resistance is not dependent on the material hardness. The contribution of the mean particle diameter of the erodent is predominant compared to the substrate conditions."
Combine Figure 6-8 and use meaningful results.
The Authors wish to thank the Reviewer for his/her constructive comment.
Fig.6-8 have been combined and the corresponding description has been amended accordingly. In addition, EDS elemental maps have been reported to corroborate the discussion of erosion behavior.
The result and discussion portion is very weak, Please rewrite the result and discussion portion and compare the results with the literature.
The Authors wish to thank the Reviewer for pointing this out.
The result and discussion section has been amended by adding further outcomes and by adding the corresponding proof to support the discussion.
Round 2
Reviewer 1 Report
Comments and Suggestions for Authors
No Comments
Reviewer 2 Report
Comments and Suggestions for Authors
The authors have revised the manuscript and answered the questions. Now, it is improved and could be accepted.
Reviewer 3 Report
Comments and Suggestions for Authors
The manuscript has been revised and acceptable in present form.